# Robots on Demand: A Democratized Robotics Research Cloud

**Victoria Dean**
Robotics Institute
Carnegie Mellon University
vdean@cmu.edu

**Yonadav Shavit**
Schmidt Futures

**Abhinav Gupta**
Robotics Institute
Carnegie Mellon University

**Abstract:** Robotics research is slowed by three challenges: building a robotics lab is expensive (few participants), everyone uses different robots (participants' findings often don't generalize outside their lab), and there is no internet-scale robotics dataset (no lab has the resources to make many robots do many different tasks to generate data and there is no data in the wild). The solution is to build a "Robotics Research Cloud" consisting of centers filled with remotely operable robots in standardized environments. This would be a valuable resource in pushing forward robot learning as a field by making cutting-edge robotics research broadly accessible, helping the field identify promising new approaches that succeed on agreed benchmarks, and creating a massive real-world robotics dataset similar to those that have revolutionized machine learning for vision and language.

**Keywords:** Remote robotics, open-source, benchmarking

## 1 Introduction

Robotics is stuck in its pre-ImageNet phase. Two major image classification competitions, the PASCAL VOC Challenge [1] and its much larger successor, the ImageNet Large Scale Visual Recognition Challenge [2], ushered in the modern era of machine learning. This was not only because of these datasets' scale but because they gave every aspiring machine learning researcher a platform on which to succeed: if you could build a program that empirically outperformed every other superstar researcher's attempt, then the field immediately paid attention to you. By expanding the number of participants in machine learning and defining a way by which they could agree on the best algorithms, ImageNet launched the modern machine learning revolution.

This transformation has not yet happened in robotics, for essentially the same reasons that plagued machine vision before ImageNet. The startup costs to creating a new robotics lab are prohibitively high: new investigators must invest not only hundreds of thousands of dollars for new robots and lab space, but also years of cumulative research hours setting up a new robot, calibrating it, and reimplementing and re-tuning numerous existing baselines just to begin contributing to the research frontier. Similarly, there is little guarantee that an algorithm achieving impressive results in one lab's setting will work well in others', due to myriad differences in robotic hardware, sensors, visual and physical properties of the testing environment, and other implementation details. As a result, even potential breakthroughs struggle to gain broader traction. Individual labs maintain their own understanding about which approaches work and build mostly on their own lab's previous work, limiting the benefits that reach the entire robotics community.

## 2 Related Efforts

Many roboticists have documented these challenges and have over the years tried to address them in different ways. Below is an overview of several such initiatives.

**Standardizing hardware** Robot platforms such as Willow Garage's PR2 and Rethink Robotics' Baxter attempt to standardize the hardware used across labs. Recent hardware efforts [3, 4, 5, 6, 7]

Blue Sky Papers, 5th Conference on Robot Learning (CoRL 2021), London, UK.

tend to involve exotic hardware targeting more niche communities. The YCB Object and Model set [8] standardizes the objects used in robotic environments. Standardized hardware is a useful step, but each robot is still only affordable to the biggest and best-funded labs, and environmental variations like lighting and sensing leave performance comparisons across labs difficult.

**Standardizing software**   ROS [9] and follow-up efforts such as PyRobot [10] provide a common software stack allowing abstraction of parts of the robot hierarchy from perception to control. However, they do not address the benchmarking or access challenges described above.

**Collecting and combining robotics data at scale**   Self-supervised experimentation (robots autonomously running experiments and evaluating their own success) allows individual labs to automatically collect large robotics datasets [11]. Google's "arm farm" further scaled data collection by using 14 robot arms working in parallel, substantially improving robotic grasping [12]. RoboNet [13], an open-source robotics dataset, facilitates data sharing across labs and has increased the scale of available robotics data. However, a dataset alone does not provide a way to easily evaluate models trained on the data or a way to compare algorithms across labs.

**Simulation benchmarking**   Many simulators provide benchmarks that standardize results in both robotics settings [14, 15, 16, 17] and embodied AI settings with navigation components [18, 19, 20]. These have the benefit of eliminating variance due to physical environment conditions across lab setups. However, real-world performance is what we care about, where sample efficiency is more important and complex methods may be impossible to tune.

**Simulation to reality**   Learning in simulation and transferring to reality is another promising approach [21, 22]. Performing most computation in simulation is an attractive way to scale robot learning, as simulation is safer and more efficient than the real world. OpenAI has shown success with sim2real, including dexterous manipulation of a Rubik's Cube [23]. However, it is next to impossible for a simulator to match the complexity and richness of the real world, making transfer inefficient. Many aspects of the real world, such as tactile sensing and dynamic interactions, are difficult to model, and learning in simulation is unlikely to yield real-world results in these domains.

**In-person robotics competitions**   Some efforts take a different approach: running in-person robotics competitions. Entrants to these challenges run their own experiments and iterate independently before all teams congregate and finally test their methods on the real environment. Notable competitions include the DARPA challenges [24, 25, 26] and the Amazon Picking Challenge [27].

**Remotely operable robotics testbeds**   The US Robotics Roadmap includes a section highlighting the need for shared and remote-access robotics testbeds [28]. Using remotely operable robots, an experimenter can deploy code from anywhere in the world and observe the results using the environment's sensors. Duckietown [29] hosts the AI Driving Olympics, a competition on a series of tasks in a simplified autonomous driving world on low-cost standardized RC cars. Participants submit software solutions remotely which are run on physical robots in the environment. As an alternative model, the Georgia Tech Robotarium [30] allows anyone to remotely access a physical robotic swarm testbed, free for academic purposes. Their swarm consists of 20 low-cost RC cars in a shared space. The Robotarium also includes a simulator for code and safety checks before physical deployment. Finally, and most similarly to our proposal, the Real Robot Challenge [31] is an annual robotic manipulation competition allowing participants to test and compare their methods remotely on real hardware (seven TriFinger robots [5]) and a corresponding simulator.

## 3   Proposed Robotics Research Cloud

These recent efforts illuminate a path forward for robotics: remotely accessible robots on which everyone can run experiments, collect data, and benchmark their algorithms' performance. On top of this, collecting and anonymously releasing the recorded trajectories would create an ever-growing corpus of open-access robot operation data, unlocking large-scale machine learning applications in the robotics realm. All that's left is to put these ingredients together: a facility full of copies of the same robot set in standardized environments, connected to the internet for all researchers to access, fundamentally accelerating robotics as a field. Below, we sketch out a proposed structure.

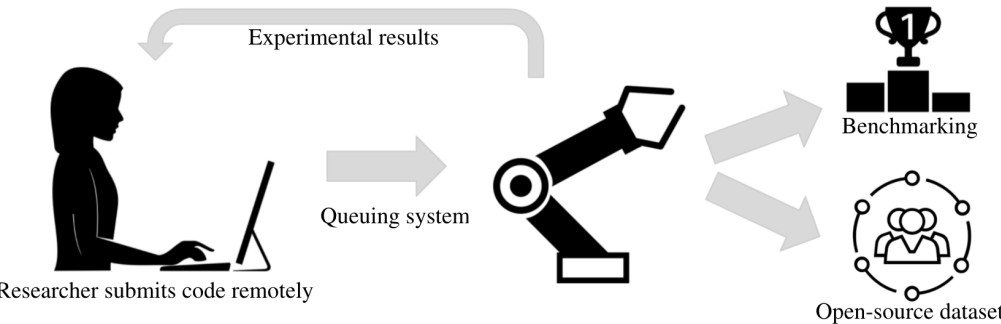

Figure 1: **Proposed workflow.** A researcher can quickly test an idea on a physical robot, obtain results, and contribute to the community through benchmarks and open-sourced data.

## 3.1 Example Workflow

A prototypical use-case is visualized in Figure 1. A researcher might start with an existing codebase, make their own tweaks, define how long they'd like this code to run for and which benchmark task it's trying to satisfy (if any), and add it to a queue. The center prioritizes and runs these experiments on real robots (after they've passed simulation checks), scores them, and uploads the sensor and operation data to the cloud where the user can access it. Depending on the amount of training time needed, the experiment turnaround time might be 24 hours. On a slower timescale, anonymized trajectories would be added to an open-source dataset for community benefit.

## 3.2 Organizational Structure

### 3.2.1 Steering Committee

The steering committee will be composed of robotics researchers and other stakeholders, responsible for decisions regarding the center's research agenda, high-level policies, and long-term development. These responsibilities would include:

- Choosing which robots and sensors to procure.

- Signing off on proposed new experiment environments and modification to existing environments based on researcher feedback.

- Determining access policies for the robots, including which institutions and researchers can run experiments and how their time is prioritized in the research queue.

- Determining the future trajectory of the Robotics Research Cloud, including whether and how to build additional centers.

### 3.2.2 Local Team

The local team will be responsible for building and maintaining the center itself. In addition to the handful of staff necessary to procure and maintain the robots, the local team may include dedicated staff necessary to reset the robots, in the experiments where environmental resets need human intervention. The local team should include a team of full-time software engineers focused on building tooling to improve the experience of the remote research community. These tools should:

- Create a unified software stack for robotic control, including a streamlined experience for remote researchers to schedule experiments, deploy code, and collect results. The codebase should allow researchers to easily take components from other research projects.

- Enable remote researchers to set up new benchmark tasks and create online leaderboards.

- Vet all code submitted to the facility against malicious content, including by maintaining a robot simulator of the robotic environment where checks must pass to ensure safety.

- Continuously publish an anonymized dataset of robot trajectories collected in the lab.

- Support a remote community through virtual discussion spaces, workshops, and tutorials.

### 3.3 Center Setup

The specifications of the first Robotics Research Cloud center should be determined through discussion and consensus among robotics researchers in the field. One possible instantiation would be a manipulation focus, as manipulation tasks are more challenging than grasping alone but more feasible than navigation or locomotion tasks, which might require more space, safety checks, and manual environment resets. Additional focus areas could be added in subsequent centers. A manipulation center could include 100 Franka robot arms, each equipped with cameras, depth and tactile sensors. These robots could have environments that enable a few benchmark tasks, such as scooping, pouring, and writing. Initial methods attempting these benchmarks could be open-sourced as out-of-the-box baselines.

## 4 Discussion

### 4.1 Where to Start

We must de-risk such an ambitious idea before building it at scale. With support from Schmidt Futures, we are prototyping a remote research setup. We aim to develop an easy-to-use interface and identify crucial features for remote experimentation. We will also gain a better understanding of staffing needs by piloting automatic safety checks and environment resets.

### 4.2 Open Questions

Many questions remain, the answers to which will impact the success of a Robotics Research Cloud.

The cloud will be most impactful if it achieves widespread adoption. Adoption depends on two key questions: Can researchers successfully run experiments remotely? Will researchers have the activation energy to adopt this new framework when many already have their own? We hope to answer the first question with our prototyping described in Section 4.1. With high-quality infrastructure and a straightforward researcher interface (which our prototype will also initiate), adoption is possible.

Choice of hardware and tasks will greatly affect research outcomes. Which sensors are necessary? How will sensor calibration and degradation be handled? Which tasks are feasible without relying heavily on humans for environment resets or benchmark evaluation?

With physical robots, safety is of paramount importance. While we can use simulation to run safety checks, how can we use an imperfect simulator to ensure safety in the real world?

Allocation of resources can be a contentious issue. How will queuing of experiments be prioritized? Should researchers be able to pay a fee to gain priority in the queue?

These questions should be discussed in detail by the robotics community to ensure success.

### 4.3 Limitations

A remote center does impose certain constraints, some of which we list here. Using fixed robots and environments rules out the possibility of jointly optimizing hardware and software. Centralizing robots also means less environment diversity as opposed to having robots individually acting in the wild. Remote teleoperation is high-latency, making collecting demonstrations or testing environment interaction difficult. Finally, latency in experimental results could increase iteration time.

### 4.4 Conclusion

By creating a facility for roboticists everywhere to run experiments and directly compare their results, we can give robotics its ImageNet breakthrough. A Robotics Research Cloud is likely to substantially accelerate the development of robotic software relative to its current trajectory. By massively increasing access to researchers across the field, it will reduce the network effects that concentrate talented robotics researchers in a small handful of schools and regions.

Next steps include securing funding and convening robotics researchers to identify the initial experimental setup and benchmarks. In the supplementary material, we present a potential timeline and budget to make the Robotics Research Cloud a reality and usher in a new age of robotics research.

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

## Potential Timeline and Milestones

Below we outline sub-goals to scale up this effort over the next 5 years.

- March 2022
  - Call for proposals for center locations
- May 2022
  - Select center location, appoint steering committee
- July 2022
  - Steering committee: produce call for proposals from robot suppliers for robot hardware and maintenance; solicit feedback from robotics community on desired environment configurations and compute
- September 2022
  - Local team: assemble core team, finalize real estate
  - Steering committee: decide details of first robot/environment/compute acquisitions
- December 2022
  - Local team: robots and environment are assembled, and locally accessible
  - Steering committee: finalize access policies and processes to include many more labs
- March 2023
  - Local team: robots are remotely accessible to a select group of remote labs, which can fly researchers to the Center for any necessary debugging. The resulting experimental data is published online.
- June 2023
  - Local team: open experiment queue to all labs meeting Steering Committee protocols
  - Steering committee: facilitate broad community adoption through workshops, challenge competitions, and tutorials
- By June 2024
  - At least 50 papers published from at least 10 universities based on center experiments
  - At least 1 proposed benchmark saturated, Steering Committee solicits calls for new environment modifications
- By June 2025
  - At least 200 papers published from at least 30 universities based on center experiments
  - Steering committee scopes out new center sited at an emerging tech hub (not an existing robotics superstar city)

## Potential Budget

Here we estimate a cost breakdown of the center for the next 10 years. Provided the first 10 years are successful, we expect interest from additional funding sources such as industry partners.

| Category | Item | Unit Cost | Volume | Subtotal |
|---|---|---|---|---|
| Capital | Robot hardware (e.g. Franka Panda) | $25000 | 100 | $2,500,000 |
| | Experiment environment (incl. sensors) | $10000 | 100 | $1,000,000 |
| | Compute + networking | $5000 | 100 | $500,000 |
| Staff | Operations staff | $75000/yr | 10 | $750,000/yr |
| | Engineering staff | $150000/yr | 5 | $750,000/yr |
| Operations | Lab space (e.g. in Pittsburgh 2020) | $25/sqft-yr | 20000 sqft | $500,000/yr |
| | Office space | $25/sqft-yr | 5000 sqft | $125,000/yr |
| | Operating expenses (electricity, furnishings, networking costs, hosting visitors) | $300k/yr | N/A | $300,000/yr |
| Total | | $3M+$2.4M per year | 10 years | $28.25M |

There are potential opportunities for savings not factored into the above estimates:

- Bulk negotiation of robots/compute could reduce costs by up to $2M (half of initial $4M).
- University-subsidized lab space could reduce costs by up to $600K a year, or $6M overall.