# OpenReview forum: "Robots on Demand: A Democratized Robotics Research Cloud"
_robot-learning.org/CoRL/2021/Conference/Blue_Sky — CoRL 2021, Blue Sky_

### Official Review · Reviewer_dEpC · 2021-08-20

**Novelty:** Poor
**Impact:** 2
**Clarity Of Presentation:** Very Good

**Recommendation:**

Strong Reject: I recommend rejecting the paper and will argue for my recommendation even if other reviewers hold a different opinion.

**Summary:**

This paper proposes a means to enabling large-scale access to standardized robotic testbeds. Specifically, it advocates for setting up centers offering remotely accessible robots. As a secondary benefit, the trajectories run on the testbed could be aggregated into large-scale datasets.

**Summary Of Recommendation:**

This paper brings up a relevant concern for the robotics community, namely, that the speed of robotics progress is hindered by lack of resources and standardization in hardware. The paper draws a valid parallel to how ImageNet has accelerated the computer vision community. It provides a complete and succinct summary of current related efforts and clearly articulates the proposed vision.

The main drawback I see in this paper is that the idea does not seem actionable for the community. The proposed robotics center involves hundreds of robots and full-time maintenance staff, which is extremely costly. The paper does not touch on possible revenue streams or funding structures. It also does not discuss how to overcome to large barrier to entry in both people and time needed to make such an effort successful. Finally, it does not outline any intermediate milestones that the community could take to work towards the larger goal.

As a result, I don't see this paper as one that has the potential to spark substantial discussion. I'm concerned that folks will generally agree that the proposed idea "would be nice", but as presented, there are no concrete steps for the community to discuss.

---

### Official Review · Reviewer_fna7 · 2021-08-27

**Novelty:** Poor
**Impact:** 3
**Clarity Of Presentation:** Very Good

**Recommendation:**

Strong Reject: I recommend rejecting the paper and will argue for my recommendation even if other reviewers hold a different opinion.

**Summary:**

This paper proposes a concept for internet-scale data collection for robotics, in order to create "a massive real-world robotics dataset similar to those that have revolutionized machine learning for vision and language." The idea is to have a center, or a number of centers, each with a large number of robots, together with technicians who can operate and maintain the robots. Remote users can then submit their research code in order to carry out experiments on these robots remotely, without having to physically be present at the center. The data from these experiments can then be shared across the world and used by other researchers.

The paper begins by motivating the need for large-scale data collection for robotics. It then describes some existing works which have attempted to facilitate this, but have failed to achieve large-scale adoption and internet-scale data. A high-level concept for the center is then proposed, which describes briefly how a remote user would be able to run experiments, although this is only limited to: "A researcher might start with an existing codebase, make their own tweaks, define how long they’d like this code to run for and which benchmark task it’s trying to satisfy". The paper also describes what the organisational structure would be, such as the need for a steering committee and a local team of staff.

The paper then finishes with some open questions about the feasibility of the proposed concept.

**Summary Of Recommendation:**

The paper is well motivated. Broadly, I agree that robotics is indeed "stuck in its pre-ImageNet phase", and that we need a significantly larger amount of data for machine learning to have a genuine impact on robotics. There is no debate about this, and it is a common understanding in the community.

So, the motivation is good. But my main concern is that the paper does not really propose a unique or interesting solution. Whilst a solution itself is proposed, it is rather an obvious solution, which can be distilled down to just having a large number of robots which are remotely accessible to researchers. This is not a surprising idea, and is essentially just a scaled-up version of existing projects that have already been attempted at a small scale. Yes, the paper describes some more details at a high level, such as the need for a steering committee, and the need to decide how much users should be charged to use the service, but these are not the important questions to be addressing at this stage. The most important questions to address are the two which the authors propose themselves: "Can researchers successfully prototype and run experiments remotely? Will researchers have the activation energy to adopt this new framework, when many labs already have their own?" It might have been better to write a paper specifically addressing these questions, rather than a paper which asks some questions which we are already aware of.

Specifically, the below issues still remain after reading the paper:


1) It is not clear how a remote user could conduct experiments without the need for a significantly expensive amount of human intervention from the center's staff. Authors state: "the local team may include dedicated staff necessary to reset the robots, in the experiments where environmental resets need human intervention." But performing experiments with robots requires much more human intervention than just resetting the environment. Experiments might require specific types of calibration, specific environment configurations, and specific ways to evaluate performance. Communication between the user and the staff would be so slow that the system would become very frustrating for the user. From my own personal experience, it takes a very long time to explain to somebody else exactly how to run the experiments I have in mind, and that is even with in-person discussions with the robot right in front of us. So generally, the paper does not address the practicalities of actually running experiments. Sentences like "The center prioritizes and runs these experiments on real robots" is about as much as we are told, but this is too vague.


2) There are very few serious robotics researchers who do not already have access to a robot. Yes, there are amateur enthusiasts and young students, but the emerging research from non-experts is not likely to yield significant gains to the professional community. Therefore, the need for running experiments remotely is not very well motivated, particularly considering the issues I raise in (1) above. We can already run experiments locally in existing research labs, and then share this data. As the authors state, staff would need to "Vet all code submitted to the facility against malicious content", and these issues such as this do not exist with local experiments. I can see something like these centers perhaps working for collecting a large number of human demonstrations, in which case there is less need for human technicians on site and the problems in (1) are less severe. But as the authors state themselves, "Remote teleoperation is high-latency, making collecting demonstrations or testing environment interaction difficult".


3) The cost of such a set of centres would be vast. Even if we could be sure that the above technical and practical issues were solved, it is not clear at all how this would be funded. No university in the world is likely to have a budget to support this. It would require significant government investment across multiple countries, and I think that the likelihood of this happening is slim. Yes, perhaps the role of the paper is to highlight these questions and to encourage the community to lobby governments and form these collaborations, but I think that this is an obvious problem which does not require a paper to highlight it. Some of the large tech companies may have budgets for something like this, but in those cases, the data would not be shared and so this would defeat the point of the paper.


4) The authors raise the issue that "there is little guarantee that an algorithm achieving impressive results in one lab’s setting will work well in others". But the idea proposed by the authors is to have "a facility full of copies of the same robot.". Whilst this solves the issue of being able to replicate results, it meanwhile raises a new issue in that any trained policies would only work on those specific robots. This would result in one robotics company having a monopoly on all of this data, since it would only be guaranteed to be valid for that company's robot, which becomes very problematic.


So in summary, whilst the motivation is clear, the solution is rather obvious and rather idealistic. In terms of practicality, the extremely large cost of setting up and maintaining these centers would be unlikely to yield a comparable return to the community. And in terms of research, the questions the paper asks are the questions we are already asking as a community, and the paper does not propose any serious answers to those questions. The paper was an interesting read, and the questions raised are important questions which no doubt we should be discussing, but I don't think that there is sufficient novelty or potential for impact for this to be "blue sky".

---

### Decision · Program_Chairs · 2021-10-01

**Decision:**

Accept

**Comment:**

The paper proposes a shared large-scale robot testbed to lower the entry barrier in robotics. The reviewers agree that the paper is well written and motivates the problem well. The reviewers are however concerned about the practicality of the proposal.

To augment the reviewers, I want to point out that the idea of the Large Scale robot testbed made its way into the US Robotics Roadmap (http://www.hichristensen.com/pdf/roadmap-2020.pdf, Pg. 79-81), which should help with funding in the upcoming years. In this AC's opinion, it is true that the obstacles are big, but the only way to get there is by having an open community-wide discussion. Blue Sky track is precisely envisioned for those discussions, and possibly a venue to gather a community around it. The benefits of having such a testbed, outweigh the difficulty -- in any area of technology, every time we made a process easier and cheaper, it made a profound difference in often unexpected ways.

Respecting reviewers opinions, I still recommend the acceptance of the paper. I recommend that the authors connect the proposal with the US Robotics Roadmap.